# snR30/U17 Small Nucleolar Ribonucleoprotein: A Critical Player during Ribosome Biogenesis

**DOI:** 10.3390/cells9102195

**Published:** 2020-09-29

**Authors:** Timothy John Vos, Ute Kothe

**Affiliations:** Alberta RNA Research and Training Institute (ARRTI), Department of Chemistry & Biochemistry, University of Lethbridge, Lethbridge, AB T1K 3M4, Canada; vost@uleth.ca

**Keywords:** rRNA processing, SSU processome, pre-ribosome, nucleolus, H/ACA RNA, endonucleolytic cleavage, RNA chaperone

## Abstract

The small nucleolar RNA snR30 (U17 in humans) plays a unique role during ribosome synthesis. Unlike most members of the H/ACA class of guide RNAs, the small nucleolar ribonucleoprotein (snoRNP) complex assembled on snR30 does not direct pseudouridylation of ribosomal RNA (rRNA), but instead snR30 is critical for 18S rRNA processing during formation of the small subunit (SSU) of the ribosome. Specifically, snR30 is essential for three pre-rRNA cleavages at the A_0_/01, A_1_/1, and A_2_/2a sites in yeast and humans, respectively. Accordingly, snR30 is the only essential H/ACA guide RNA in yeast. Here, we summarize our current knowledge about the interactions and functions of snR30, discuss what remains to be elucidated, and present two non-exclusive hypotheses on the possible molecular function of snR30 during ribosome biogenesis. First, snR30 might be responsible for recruiting other proteins including endonucleases to the SSU processome. Second, snR30 may contribute to the refolding of pre-rRNA into a required conformation that serves as a checkpoint during ribosome biogenesis facilitating pre-rRNA cleavage. In both scenarios, the snR30 snoRNP may have scaffolding and RNA chaperoning activity. In conclusion, the snR30 snoRNP is a crucial player with an unknown molecular mechanism during ribosome synthesis, posing many interesting future research questions.

## 1. Introduction

Eukaryotic ribosomes are synthesized in the nucleolus with the help of hundreds of assembly factors. In a highly orchestrated fashion, multiple protein and RNA assembly factors are recruited to nascent pre-ribosomal RNA (pre-rRNA) and depart at defined times, whereas ribosomal proteins also associate in a defined order, eventually forming ribosomal subunits together with the matured rRNAs. Among the ribosome assembly factors, the nucleolar snR30 is an essential H/ACA RNA that is critical for the formation of the small ribosomal subunit, but only limited information is available on the mechanistic role of snR30 during the complex process of ribosome formation [1]. Before discussing the function of snR30 in detail, we will first provide a brief overview of eukaryotic ribosome synthesis, in particular, small subunit (SSU) formation, to place the role of snR30 into context. Unless indicated, the presented information is based on the *Saccharomyces cerevisiae* model system used widely in ribosome biogenesis research (as reviewed in References [2,3,4]).

Ribosome biogenesis occurs co-transcriptionally on the pre-rRNAs, and snR30 is also thought to act co-transcriptionally during the processing of 18S rRNA. Three of the four mature rRNAs (18S, 5.8S, and 25S rRNA) are transcribed by RNA polymerase I as a single 35S precursor rRNA (pre-rRNA, 47S pre-rRNA in humans), whereas the fourth rRNA, 5S rRNA, is transcribed independently by RNA Polymerase III. In addition to the mature rRNA sequences, the 35S pre-rRNA contains two external and two internal transcribed spacers abbreviated as ETS and ITS respectively, which are removed through several nucleolytic cleavage events such that these regions are not present in the mature ribosome. The initial site of pre-rRNA cleavage, that separates the 18S rRNA from the remaining pre-rRNA, are called A sites, specifically A_0_, A_1_, A_2_, and A_3_ in yeast and 01, 1, and 2 in humans (Figure 1) [5,6,7]. Notably, snR30 is involved in pre-rRNA cleavage at sites A_0_, A_1_, and A_2_.

During ribosome biogenesis, snR30 does not act alone, but rather forms a stable ribonucleoprotein (RNP) that belongs to the family of H/ACA small nucleolar RNPs (snoRNPs). The snoRNPs represent a large portion of the ribosome biogenesis factors found in the nucleolus, and most snoRNPs chemically modify rRNA, but notably, this is not the case for the snR30 snoRNP. Occurring in the form of either C/D box or H/ACA box guide RNA systems, snoRNPs direct site-specific 2′-O-methylation and pseudouridylation, respectively [8,9]. Each of these complexes is composed of one guide RNA, also called small nucleolar RNA (snoRNA), and four core proteins. The proteins Nop1 (Fibrillarin in vertebrates), Nop56, Nop58, and Snu13 (15.5K) assemble on C/D box guide RNAs, forming C/D snoRNPs [10,11,12,13], whereas the proteins Cbf5 (Dyskerin), Nop10, Gar1, and Nhp2 are components of the H/ACA box snoRNP together with H/ACA box guide RNA [14,15,16]. During ribosome assembly, the H/ACA snoRNPs introduce many pseudouridines in rRNA which play an important role in the maintenance of ribosome structure, stability, and translational fidelity [17,18,19,20,21,22,23].

Interestingly, of the more than one hundred snoRNAs known in yeast, only three guide RNAs are essential, namely U3, U14, and snR30. In addition, deletion of *S. cerevisiae* snR10 results in slow growth, and a cold-sensitive phenotype [24]. Remarkably, the primary function of these essential snoRNAs is not rRNA modification, but they are all directly or indirectly required for rRNA processing during ribosome synthesis. Additionally, snR30 may also play a role in cholesterol trafficking in higher eukaryotes [25,26,27]. In comparison to snR30, much more is known about the essential U3 snoRNA, a C/D box RNA that is responsible for the correct folding of the central pseudoknot in the 18S rRNA [28]. The central pseudoknot is an SSU rRNA structure conserved from prokaryotes to higher eukaryotes that connects the different domains of the SSU rRNA [29]. To facilitate pseudoknot formation, U3 makes multiple essential base pair interactions to the pre-rRNA in both the 5′ ETS and the 18S rRNA [28]. In particular, the box A and A′ motifs in the U3 snoRNA bind to the 18S rRNA regions which form the central pseudoknot. When U3 is deleted, biogenesis of the small subunit is halted early due to the U3 snoRNP having a central role in biogenesis. This leads to accumulation of 23S pre-rRNA due to an inability to process at A_0_, A_1_, or A_2_ [30]. Lastly, U14 is another C/D RNA that is essential for 18S rRNA formation [31]. U14 appears to have two distinct roles, as one region, that base pairs to the pre-rRNA and is not essential, guides the introduction of a 2′-O-methylation at C414 in 18S rRNA, whereas another region is essential and binds to the 18S rRNA extensively in the 5′ domain of the 18S rRNA [32,33,34]. Depletion of U14’s essential region leads to the exact same phenotype as depletion of U3 or snR30. Thereby, U14 resembles the H/ACA box RNA snR10 which also has dual functions by directing pseudouridine formation at position 2923 in 25S rRNA and by contributing to 18S rRNA processing. In cases where snR10 is lacking, the phenotype is similar to a knockout of the essential genes, except less pronounced. There is minor 35S pre-rRNA accumulation that leads to an increase in 23S pre-rRNA and a 21S pre-rRNA product. However, at permissive temperatures, this pre-rRNA is still processed, albeit slowly, into mature 18S rRNA [35].

Knowledge of ribosome biogenesis, and specifically small subunit biogenesis, has significantly increased due to recent advances in cryo-electron microscopy (cryo-EM), yielding high-resolution structures of the SSU processome complex [36,37,38], but no structural insight of snR30 bound to the SSU processome is available to date. The recent cryo-EM structures provide insight into the interactions within the SSU processome which has led to advances in both the understanding of protein positioning and timings of large-scale rearrangements in the small ribosomal subunit during ribosome biogenesis. However, many of the early-acting factors, such as the snR30 snoRNP, are not present in the structures solved so far, presumably because the snR30 snoRNP has already dissociated before formation of the stable SSU processome. Therefore, no structural information is currently available on the interaction of snR30 with the SSU processome, but instead, our knowledge is derived mostly from genetic and biochemical studies. For example, by tagging and purifying a known biogenesis factor, the timing of SSU processome association of this factor can be roughly estimated by identifying other co-purified factors [39,40]. More recently, different ribosomal precursors have also been studied by expressing and purifying truncated and tagged pre-rRNA variants followed by proteomic analysis, providing additional information on when assembly factors associate and dissociate during pre-rRNA transcription [41,42,43,44]. Notably, taking such an approach, snR30 was detected in early ribosomal intermediates that arise before 18S rRNA transcription is complete, but it was no longer detected in later intermediates [41]. These findings suggest that snR30 acts relatively early during ribosome biogenesis when the structure of the SSU processome might still be rather flexible and labile.

The early stages of eukaryotic ribosome synthesis are characterized by a dynamic interaction network of several factors, including snR30. Ribosome biogenesis starts with the binding of the UTP-A complex to the 5′ ETS of the 35S pre-rRNA transcript. Subsequently, the U3 snRNP, UTP-B, and Mpp10 complexes associate [41,42,45]. While these complexes contain a large portion of the assembly factors, there are many individual proteins, small complexes, and modification enzymes which also act during this period. snR30 is part of one of these complexes that seems to only be present until the 3′ minor domain of 18S rRNA is transcribed [41]. The primary interaction point of snR30 is within expansion segment 6 (ES6) in the central domain, which is unstructured and therefore not resolved in the cryo-EM structures of the SSU processome [38]. The possible function of snR30 binding for pre-rRNA folding will be further discussed below.

By recruiting all these assembly factors, the pre-rRNA in the SSU processome is held in a state where the mature subunit is recognizable, but not fully formed, as evident in the cryo-EM structures [36,37,38], and snR30 may contribute to preventing premature interactions as further discussed below. One of the most striking differences between the SSU processome structure compared to the mature 40S subunit is that the central pseudoknot is unable to form due to the presence of Sas10, Lcp5, the U3 snRNP, and other factors [37]. Furthermore, the SSU processome separates the four domains of the 18S rRNA into independent units which can fold separately [38,46]. More recent structural information from the thermophilic fungus *Chaetomium thermophilum* has suggested that the 5′, central, 3′ major, and 3′ minor domains of 18S rRNA are not sequentially integrated into the pre-40S subunit, but that the 5′ domain may join the SSU processome last; however, it remains to be investigated whether this finding holds true in other organisms [47]. Thus, the early stages of SSU formation are characterized by dynamic interactions and several conformational changes in rRNA which are facilitated by many factors, including snR30.

In summary, research into ribosome biogenesis is a rapidly evolving field as the concerted functions of hundreds of poorly understood factors and steps including snR30 need to be identified. In this review, we consolidate the information known about snR30 and develop hypotheses about its role during ribosome formation. Thereby, we hope to inspire future research into the function and mechanism of snR30 which will fill a critical gap in our understanding of ribosome formation.

## 2. snR30 Biogenesis, Sequence, and Structure

### 2.1. Transcription and Processing

In unicellular eukaryotes, such as *S. cerevisiae*, *snR30* is transcribed as an independent gene by RNA Pol II, the mRNA producing polymerase. Contrary to mRNA, the snoRNA undergoes processing to remove the polyA tail by initially binding the Nrd1-Nab3-Sen1 complex which presents the snR30 3′ end to the exosome for exonucleolytic processing [48,49]. On the 5′ end, snR30 possesses a trimethyl guanosine cap like most other H/ACA snoRNAs in yeast [25,50], which is added by the conserved methyltransferase Tgs1 [51].

In multicellular eukaryotes such as humans, transcription and processing of U17a and U17b, two human homologs of snR30, is significantly different. Both U17a and U17b are encoded by the *U17HG* gene upstream of the cell cycle regulatory gene *RCC1* [52]. Interestingly, the *U17HG* gene harbors *U17a* and *U17b* in two introns; however, the *U17HG* gene seems not to encode a protein and only the *U17* sequences are conserved [53]. However, in other species, the location of the *U17* gene varies such as in *X. laevis*, where *U17* is transcribed in all six introns of the r-protein *S8* gene [54]. Following transcription of the host gene, the excised intron containing U17 is then processed exonucleolytically at both the 5′ and 3′ ends in both Xenopus and HeLa cells, implying a conserved mode of maturation [55,56]. The 3′ end is processed by the exosome [57], whereas the 5′ end is processed by an unknown endonuclease [55,58]. Further information on snoRNA biogenesis and processing events have been reviewed by Kufel and Grzechnik [59].

### 2.2. snR30 snoRNP Formation and Nucleolar Localization

In yeast, the core snR30 RNP comprising the RNA and the four canonical H/ACA proteins is formed in the same way as other H/ACA snoRNPs that modify rRNA. This process begins with formation of a protein complex initiated by the binding of the protein Shq1 to Cbf5 in the cytoplasm [60]. Shq1 associates with the RNA-binding domain of Cbf5 mimicking interactions of H/ACA snoRNA, which is known to be strongly bound by Cbf5 [61,62]. The Shq1-Cbf5 complex is then shuttled into the nucleus where the majority of RNP maturation takes place. During this process, Nop10, Nhp2, and an assembly factor called Naf1 bind, forming a large protein complex [63]. At the site of snR30 transcription, a pair of ATPases catalyzes the release of Cbf5 from Shq1, freeing the enzyme to tightly bind the snoRNA in its place [64]. This interaction is mediated by Naf1 binding to the large subunit of RNA polymerase II [65]. After binding of Cbf5 to the H/ACA guide RNA, the subsequent maturation step involves the competitive binding of Gar1 to Cbf5, resulting in the displacement of Naf1 which is recycled back to the cytoplasm for further maturation. The now fully assembled H/ACA snoRNP is shuttled from Cajal bodies to the nucleolus by Nopp140 [63,66]. Localization signals for transport into the nucleolus appear to be located in the H and ACA boxes, as well as the general structure of the guide RNA, at least for vertebrates [67,68]. For U17, the ACA box is required for assembly in vitro and presumably in vivo, signifying that nucleolar localization is dependent upon formation of the RNP [69]. As only mature snoRNPs are found within the nucleolus, maturation must occur between transcription in the nucleoplasm and import into the nucleolus [70]. It is possible that some steps of snR30 maturation may occur within Cajal bodies similar to the maturation of other snoRNAs. Immature C/D box RNPs as well as a subgroup of H/ACA RNAs called H/ACA small Cajal body RNAs (scaRNA) can be readily detected within Cajal bodies, and some evidence suggests that H/ACA snoRNPs may in general also traverse Cajal bodies [71,72,73]. In conclusion, current evidence suggests that the maturation of the snR30/U17 RNP follows the same steps as canonical H/ACA guide RNA as all H/ACA RNAs including snR30 assemble with the same proteins.

### 2.3. Conservation of snR30/U17 Structure, Sequence, and Motifs

Typical H/ACA guide RNAs in eukaryotes share a similar secondary structure comprised of two hairpins connected by a hinge region. The two hairpins are followed by two conserved sequence motifs, the H Box (ANANNA) and the ACA Box, respectively. While most known H/ACA RNAs contain two hairpins in eukaryotes, there are instances of H/ACA guide RNAs having one or three hairpins in selected eukaryotic organisms as well as in archaea. snR30 is an unusual H/ACA guide RNA that has two primary hairpins, the 5′ and 3′ hairpins, but also possesses a third internal hairpin as well as a 41-nt leader sequence at its 5′ end (Figure 2). Notably, the 5′ hairpin of snR30 is much longer than a standard H/ACA hairpin such that *S. cerevisiae* snR30 has an unusual length of 606 nt, almost triple the length of the average yeast H/ACA guide RNA (~200 nt). While not found in all H/ACA snoRNAs, the internal hairpin and the 41-nt leader are both features that are also present in some other H/ACA guide RNAs [74]. Interestingly, snR30 lacks an unpaired internal bulge following the first stem of the 5′ hairpin, a feature known as the pseudouridylation pocket in standard H/ACA guide RNAs. In contrast, the 3′ hairpin contains a single-stranded bulge like all other H/ACA guide RNA hairpins, and the top of the bulge is located at a conserved 14–16 nucleotide distance from the base of the hairpin and the ACA box [75]. In H/ACA snoRNAs directing pseudouridylation, this distance is important for properly positioning the guide RNA on the Cbf5-Nop10-Nhp2 binding surface, allowing binding of the target RNA to the pseudouridylation pocket and positioning of the target uridine into the active site of Cbf5 [62,75]. In the 5′ hairpin of snR30, the only similar bulge occurs too far away from the base of the stem and the H box for correct positioning of Cbf5. Accordingly, no pseudouridine has been suggested to be introduced by the 5′ hairpin of snR30. Similar to canonical H/ACA snoRNAs and based on the location of the H and ACA Boxes, snR30 is expected to bind one set of the H/ACA core proteins (Cbf5, Nop10, Gar1, and Nhp2) to each of the 5′ and 3′ hairpins, resulting in a predicted 2:1 stoichiometry between the proteins and the snR30 RNA. Hence, despite its elongated structure, the only unique aspect of snR30 compared to modification H/ACA snoRNAs is that its 5′ hairpin does not have any known RNA targets.

Human U17 RNA (207 nt) is shorter than yeast snR30 and comprises four hairpins (Figure 3B), forming a secondary structure consisting of a 5′-variable domain and a 3′-conserved domain [25,74,76]. Thus, the comparison of yeast snR30 and human U17 can reveal functionally important regions of this conserved H/ACA snoRNA. During evolution, the 5′ region of snR30/U17 was compacted to a smaller size, effectively reducing transcriptional cost, which is similar to the general trend of guide RNA shortening between single-cell and complex eukaryotes [77]. In humans, the 5′ end of U17 is composed of two stems of similar size prior to the H box. Whereas the H box of U17 is not required for in vitro RNP formation, the H box of snR30 is critical for accumulation of snR30 in vivo [69,74]. U17 also contains an internal hairpin, although it is much smaller than that of snR30. Since the 5′ structure of snR30/U17 is not conserved, mutational studies investigated whether the 5′ and internal hairpins of yeast snR30 are critical for cell viability [74]. Indeed, both the 5′ hairpin and the internal hairpin can be individually deleted without affecting cell viability, and cell viability was maintained at a reduced level when replacing both the 5′ and internal hairpins with the 5′ hairpin of another box H/ACA RNA. Therefore, the 5′ hairpin and the internal hairpin of yeast snR30 are not directly responsible for its essential role within the cell. In contrast to the 5′ domain, the 3′ hairpin of U17 is extremely similar to that of snR30 as they both possess a structure identical to a standard H/ACA guide RNA hairpin including an unpaired bulge. As further outlined below, the conserved nature of the 3′ hairpin already indicates that this region in snR30/U17 is functionally most important.

By aligning the sequences of the snR30/U17 species from yeast, *Xenopus*, and humans, two strongly conserved sequence motifs in the 3′ hairpin were discovered and called m1 and m2, which are critical for ribosome formation [74]. The m1 and m2 regions are located in the non-productive, unpaired ‘pseudouridylation pocket’ on the 3′ hairpin of snR30/U17. However, rather than being located on the distal side of the pocket where modification H/ACA guide RNAs bind their targets, they are located on the basal side of the pocket. Two complementary sequences in 18S rRNA were discovered and designated as rm1 and rm2, and mutational studies confirmed Watson-crick base pairing between the m1/rm1 and m2/rm2 sequences that is necessary for pre-rRNA processing [74]. While there is some variation in the m1 and m2 sequences across eukaryotes, these are always matched by compensatory mutations in the rm1 and rm2 motifs in 18S rRNA (Table 1), underlining the importance of this base-pairing of snR30/U17 with 18S rRNA for ribosome biogenesis.

**Figure 2 cells-09-02195-f002:**
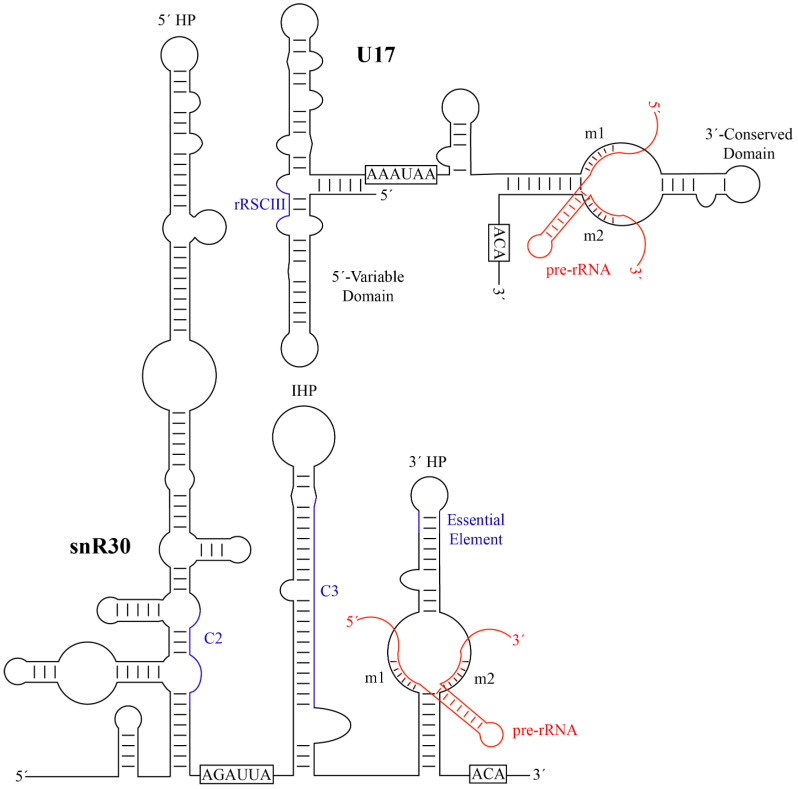
Secondary structures of yeast snR30 and human U17 and their modus operandi of pre-RNA binding. The H and ACA sequences, that characterize each H/ACA snoRNA, are boxed. The m1 and m2 motifs are labelled and the base-pairing to target pre-rRNA is shown in red. Additional regions of snR30/U17 predicted to have a function such as forming further interactions with 18S rRNA are depicted in blue (C2 and C3 in *S. cerevisiae* snR30, rRSCIII in human U17). HP, hairpin; IHP, internal hairpin. Minor bulges and imperfect base-pairing are not represented. snR30 was adapted from Atzorn et al. [74], and U17 was adapted from Ruhl et al. [78].

In addition to the critical and highly conserved m1 and m2 regions, additional elements of snR30 or U17 have been identified that also bind to 18S rRNA, although these secondary interactions are not conserved across all species. In *S. cerevisiae*, crosslinking, ligation, and sequencing of hybrids (CLASH) uncovered additional areas of interaction between snR30 and 18S rRNA [34]. Notably, two of the strongest interaction sites, called C2 and C3, also contain regions of sequence complementarity between snR30 and 18S rRNA and are conserved among fungi (Figure 3). First, a region in the 5′ hairpin of snR30 is proposed to interact with helix 1 in expansion segment 6 (ES6) of 18S rRNA in close vicinity to the interaction of the m1 and m2 regions with helix 3 of ES6. However, the importance of this interaction remains to be investigated since the 5′ hairpin of snR30 is dispensable. Second, a 19-nt region within the internal hairpin of snR30 has the potential to base-pair to expansion segment 7 (ES7) in 18S rRNA, which is also conserved in *Schizosaccharomyces pombe*. Whereas these interactions sites are likely specific to fungi, other putative contacts between U17 and 18S rRNA have been reported in vertebrates. In humans, the U17 rRCSIII sequence in stem 2 of the 5′ domain is predicted to base-pair with 18S rRNA at positions 967–976 (Figure 2) [79], and this sequence complementarity is conserved not only in mammals, but also birds, fish, amphibians, and reptiles. An additional element in stem 1 of the 5′ domain, called rRCSI, may form 12 base-pairs to the 18S rRNA, but is only conserved in fish and amphibians, not in mammals [79]. It remains unknown whether these predicted contacts between vertebrate U17 and 18S rRNA form in vivo and whether they are of functional importance for ribosome biogenesis. Due to the divergence of snR30/U17 sequence and structure over evolution, different interactions with 18S rRNA may have formed that may serve similar functions in stabilizing binding of this H/ACA snoRNA to the SSU processome.

**Table 1 cells-09-02195-t001:** Comparison of snR30/U17 RNA across eukaryotes.

Organism	Common Name	Gene Name	Length (nt)	m1	m2	Accession Number	Reference
*Apteryx rowi*	Kiwi	*E1/U17*	205	AUAUUCCUA	AAACCAU	XR_003255265.1	N/A
*Caenorhabditis elegans*	Round worm	*CeN96*	221	AUAUUCCUU	AAACCAU	AY948718.1	[80]
*Caretta caretta*	Turtle	*U17*	213	AUAUUCCUA	AAACCAU	AJ306558	[79]
*Candida glabrata*	Yeast	*snR30*	576	AUAUUCCUG	AAACCAU	URS00006E7B8C	[81] ^a^
*Danio rerio*	Zebra fish	*E1/U17*	>140	AUAUUCCUA	AAACCAU	LR812563.1	N/A
*Fugu rubripes*	Pufferfish	*U17*	218	AUAUUCCUA	AAACCAU	X94942	[82]
*Homo sapiens*	Humans	*U17*	207	AUAUUCCUA	AAACCAU	L16791	[74]
*Kazachstania naganishii*	Yeast	*snR30*	589	AUAUUCCUA	AAACCAU	URS0000BE6F45	[81] ^a^
*Mus musculus*	Mouse	*U17/SNORA73*	203	AUAUUCCUA	AAACCAU	XR_003836540.1	N/A
*Pelomedusa subrufa*	Turtle	*U17*	215	AUAUUCCUA	AAACCAU ^c^	AJ306565	[79]
*Saccharomyces cerevisiae*	Yeast	*snR30*	606	AUAUUCCUA	AAACCAU	NR_132204	[74]
*Saccharomyces pombe*	Yeast	*U17*	325	AUAUUCCUA	AAACCAU	AJ544685	[74]
*Tetrahymena thermophila*	Algae	*U17*	~240 ^b^	AUAUUCCUG	AAACCAU	AJ544686	[74]
*Xenopus laevis*	Frog	*U17*	222	AUAUUCCUA	AAACCAU	X71081	[54]

Non-conserved nucleotides in the sequences from *C. glabrata* and *T. thermophila* are highlighted in bold. Accession numbers are from GenBank except for *S cerevisiae*, *K. naganishii*, and *C. glabrata*, which are from NCBI or RNA Central. References refer to the publication in which each motif was identified; no reference is provided, indicated as N/A, when the RNA sequence was only identified through a BLAST search at NCBI. ^a^ The motif was found manually in the sequences retrieved from RNA Central. ^b^ Accession number provides only partial sequence. Atzorn et al. [74] purified and sequenced the 5′ end of *T. thermophila* snR30, and its length was estimated based on gel size. ^c^ The *Psu* sequence has two A insertions 5′ of the motif, however, the rRNA motif stays the same, indicating that the insertion does not affect base pairing.

## 3. Role of snR30/U17 in rRNA Processing

### 3.1. Site-Specific Binding to the 18S rRNA

To promote formation of the small ribosomal subunit, snR30/U17 binds to the pre-rRNA and enables cleavages at sites A_0_, A_1_, and A_2_ (Figure 2) [26,83]. However, the molecular mechanism of the snR30/U17 snoRNP, including the details of its interaction with the SSU processome, remain unknown. As mentioned, snR30/U17 base-pairs to expansion segment 6 of the 18S rRNA with the m1 and m2 motifs, which flank both sides of the basal end of the pocket on the 3′ hairpin. This snoRNA–rRNA interaction is markedly different from rRNA recognition by modification H/ACA RNAs, which bind to their target using the distal half of the pseudouridylation pocket. Therefore, it is this inverse binding from snR30/U17 that prevents it from introducing a pseudouridine into the 18S rRNA (Figure 3) [74]. It has been speculated that this change in orientation allows for snR30 to stay on the SSU processome longer, until it is specifically removed by an external factor. Notably, the base-pairing interaction between m1/rm1 and m2/rm2 is essential, indicating that the binding of snR30’s 5′ hairpin to ES6 of the rRNA is of functional importance [84]. Binding of snR30 to ES6 sequesters the rm1 and rm2 regions of the pre-rRNA and the sequence between rm1 and rm2 is predicted to form a new hairpin structure in ES6 that is not present in the mature 18S rRNA, as observed in the ribosome (Figure 3) [84]. At the same time, formation of this new helix would result in the unwinding of ES6 hairpin 3 (ES6H3) and possibly also ES6 hairpin 2 (ES6H2). In addition to the interaction of snR30 with the rm1 and rm2 sequences in 18S rRNA, other interactions between the snR30 snoRNP and the SSU processome will likely occur and stabilize the interaction, but these interactions might differ between organisms. In fungi, snR30 was observed to crosslink to the C2 and C3 sites which are in relatively close proximity to the rm1 and rm2 sequences (also called C1) in the secondary structure of 18S rRNA (Figure 3A) [34]. The ES6 structure is not resolved in the early SSU processome structures when snR30 is expected to bind, but recent structural information on the 90S pre-ribosome to 40S subunit transition in *C. thermophilum* suggest that this region becomes further organized during cleavage of the A1 site [85]. These later intermediates indicate that the C1, C2, and C3 interaction sites of *S. cerevisiae* snR30 with 18S rRNA are located in close three-dimensional proximity on the surface of the early SSU processome, allowing simultaneous interaction with the snR30 snoRNP. snR30 alters the secondary structure of 18S rRNA in ES6, but the functional importance of this conformational change is unknown. By preventing the 18S rRNA from adopting its mature conformation, snR30 may hold the SSU processome in a higher energy state until additional factors cause the release of snR30 from the SSU processome (vide infra). Interestingly, a similar restructuring of the pre-rRNA is also seen upon U3 binding to the 5′-ETS region of the pre-rRNA, promoting formation of the central pseudoknot. Therefore, the essential snoRNAs (snR30, U3, U14, as well as snR10) may generally bind to the transcribing pre-rRNA acting as rRNA chaperones by keeping the rRNA in a particular conformation to avoid misfolding of the rRNA [28]. The change in ES6 conformation induced by snR30 could be a critical signal transmitted through the rRNA to sites A_0_, A_1_, and A_2_, allowing the SSU processome to surpass a checkpoint such that the next step along the maturation pathway, namely pre-rRNA processing, may occur.

Another possibility is that the snR30-induced conformational change in rRNA influences the recruitment of protein assembly factors. Certain factors may only bind to the particular rRNA conformation induced by snR30 and will thus depend on snR30 association with the ribosome. Other proteins may be unable to bind to this snR30-induced rRNA conformation and will therefore depend on snR30 dissociation. Thus, the interaction of snR30 with pre-rRNA may regulate the timing of protein association with the SSU processome.

### 3.2. snR30 Protein Recruitment

Since snR30 has no endonuclease activity itself, it must indirectly promote pre-rRNA cleavage events during ribosome biogenesis. To understand the molecular mechanism of snR30, it is therefore mandatory to dissect the interaction network of snR30 not only with rRNA, but also with additional ribosome assembly factors, including putative endonucleases which may be recruited by snR30. Notably, the endonuclease(s) responsible for rRNA cleavage at the A_0_, A_1_, and A_2_ sites have not been unambiguously identified so far. The PIN endonuclease Utp24 has been proposed to catalyze the cleavages at sites A_1_ and A_2_ [87,88], but Rcl1 also has the ability to cleave A_2_ in vitro [89].

Several assembly factors have been identified as interaction partners of snR30 through immunoprecipitations (Figure 4). Besides the H/ACA core proteins Cbf5, Nop10, Gar1, and Nhp2, snR30 also binds to Nop6 [83,90], the DEAD-box helicases Has1 [91] and Rok1, the PIN domain endonuclease Utp23, and Kri1 [92,93]. Additional interactions of snR30 were reported with ribosomal proteins S9 and S18, and the histones H2B and H4, but these may possibly represent unspecific interactions [83,90].

The interaction of snR30 with Utp23 is of particular interest as snR30 and Utp23 together may serve as an essential assembly platform to facilitate pre-rRNA processing and ribosome formation. The essential 3′ hairpin and the internal hairpin of snR30 strongly crosslink to the protein Utp23, which also directly interacts with the H/ACA core protein Nhp2 [87]. In addition to the core components of the snR30 snoRNP, Utp23 also binds Rok1, Rrp7, and Utp24 in both yeast and human, underlining the role of Utp23 as a critical assembly hub during ribosome biogenesis [87]. Whereas snR30 is required for the incorporation of both Utp23 and Kri1 into the pre-ribosome, interestingly, Utp23 is in turn needed to later release snR30 from the pre-ribosome [93]. Like Utp24, a putative rRNA endonuclease, the snR30-interaction partner Utp23 also contains a PIN domain endonuclease fold. However, the functional importance of this domain remains unsolved as yeast Utp23 does not possess the catalytic residues required for endonucleolytic cleavage [94]. In contrast, human Utp23 (hUtp23) contains catalytic residues and these are essential for cell viability, suggesting that Utp23 may possibly play additional roles in human ribosome formation compared to yeast [87]. Obviously, the direct interaction of Utp23 with the catalytically active endonuclease Utp24 may be responsible for mediating the role of snR30 in facilitating pre-rRNA processing. In this context, it is important to note that Utp23 will not simply recruit Utp24 to the pre-ribosome as Utp24 is known to interact early with the pre-ribosome already while the 5′-ETS is being transcribed [95]. Therefore, the recruitment of the snR30 snoRNP together with Utp23 to the pre-ribosome is likely followed by the subsequent interaction of Utp23 and Utp24 on the pre-ribosome, which could lead to a re-positioning of Utp24 within the pre-ribosome. Since Utp24 strongly crosslinks and directly interacts with the U3 snoRNA [88], further conformational changes in the pre-ribosome could be indirectly induced by snR30 and Utp23. Thus, snR30, Utp23, and Utp24 could be critical in coordinating conformational changes across rRNA domains within the three-dimensional (3D) structure of the pre-ribosome leading to pre-rRNA cleavage at sites A_0_, A_1_, and A_2_ by Utp24, Rcl1, or an unknown endonuclease.

### 3.3. Hypothetical Functions of snR30 during Ribosome Formation

As outlined above, snR30 acts as a critical assembly hub during ribosome biogenesis by interacting both with 18S rRNA and by binding critical assembly factors. Both through the snR30-rRNA (Figure 3) as well as the snR30–protein interactions and networks (Figure 4), snR30 can directly or indirectly facilitate processing of pre-rRNA. Accordingly, we propose the following two hypotheses regarding the molecular mechanism of snR30, which are also summarized in Figure 5.

#### 3.3.1. snR30 Mechanism—Hypothesis 1

Upon binding of snR30 with its 3′ hairpin to the expansion segment 6 (ES6) of 18S rRNA, snR30 induces conformational changes in the pre-rRNA, acting as an rRNA chaperone. These conformational changes caused by snR30 may include both the formation of specific structures such as the new helix in ES6 (Figure 3) as well as the unfolding of prematurely formed rRNA structures. The snR30-induced changes in pre-rRNA may be either transmitted directly as rRNA conformational changes through the SSU processome or may indirectly enable the recruitment of additional ribosome assembly factors to the snR30-induced conformation of pre-rRNA, leading to the correct positioning of the pre-rRNA cleavage sites A_0_, A_1_, and A_2_ relative to the responsible endonucleases, such that snR30 enables pre-rRNA processing.

#### 3.3.2. snR30 Mechanism—Hypothesis 2

The snR30 snoRNP strongly interacts with Utp23 through direct RNA–protein interaction as well as Nhp2-Utp23 protein–protein interaction, thereby recruiting Utp23 to the SSU processome, where Utp23 coordinates the binding and stabilization of additional ribosome assembly factors such as Rok1 and Kri1. Most importantly, Utp23 will interact with the SSU processome-bound PIN domain endonuclease Utp24, possibly inducing conformational changes in Utp24 and the SSU processome, enabling processing of pre-rRNA at sites A_0_, A_1_, and A_2_.

Importantly, these two hypotheses are not mutually exclusive, and it can rather be envisioned that both mechanisms occur during ribosome biogenesis, leading to concerted conformational changes in both pre-rRNA and protein assembly factors that enable pre-rRNA processing.

## 4. snR30 Release from the Pre-Ribosomal Particle

### 4.1. Required Factors for snR30 Release

Unlike canonical H/ACA snoRNPs that modify rRNA, the snR30 snoRNP requires RNA helicases to catalyze snR30 release from the pre-rRNA [91]. The dependence of snR30 on a helicase may result from the fact that the rRNA is bound in the inverse orientation to the internal bulge in the 3′ hairpin of snR30 compared to substrate RNA binding by canonical H/ACA snoRNAs. As a consequence of this inverse orientation, no pseudouridine can be formed by the snR30 snoRNP. In canonical H/ACA snoRNAs, pseudouridine formation triggers a conformational change in Gar1 which in turn catalyzes substrate turnover by altering the conformation of the thumb loop in Cbf5 which interacts with the target RNA [96]. Thus, the helicases may replace Gar1’s function in facilitating dissociation of the snR30 snoRNP from pre-rRNA.

The DEAD-box helicase Rok1 is a critical player in snR30 release from the SSU processome [92], but the helicase Has1 as well as Utp23 are also essential for this function [91,93]. Rather than directly removing snR30 from the ribosome, the helicase Rok1 is involved in a complex interplay with another critical ribosome assembly factor, the protein Rrp5, and together, Rok1, Has1, Utp23, and Rrp5 contribute to snR30 dissociation from the pre-40S ribosome. Similar to snR30, Rok1, Utp23, and Kri1, the assembly factor Rrp5 is recruited to the SSU processome during transcription of the central domain and acts as a compaction factor [97]. Unlike most other ribosome assembly factors, Rrp5 is important for the maturation of both the small subunit and the large subunit. Absence of Rrp5 prevents pre-rRNA cleavages at the A_0_, A_1_, and A_2_ sites, like depletion of snR30, but also abolishes cleavage at the A_3_ site by the ribonuclease MRP [98]. Rok1 directly binds to the A_2_ site preventing early processing, and after domain 1 of the 25S rRNA is transcribed, the protein complex Noc1/Noc2 rearranges Rrp5, thereby freeing the A_2_ cleavage site. At this time, Rok1 catalyzes the release of Rrp5 from the SSU processome [99,100]. Interestingly, the release of Rrp5 from the pre-40S ribosome by Rok1 is a prerequisite for snR30 dissociation. Thus, the most likely scenario is that Rok1 first induces conformational changes in Rrp5 or rRNA causing its release, and subsequently, the snR30 snoRNP is actively removed from the SSU processome with the help of the helicase Has1. The removal of snR30 may further be mediated by the direct interaction of Rrp5 with Has1 [100], but the molecular details of the interaction network of Rok1, Rrp5, Has1, and snR30 remain to be elucidated.

In addition to the helicases Rok1 and Has1, the PIN endonuclease Utp23 is also required for snR30 release [93]. Since snR30 is required for Utp23 binding in the first place, it is currently not clear how Utp23 is required for snR30’s dissociation [93]. Possibly, Utp23 is not directly required for the snR30 release, but rather for a previous functional step of snR30 such as mediating pre-rRNA processing (vide supra). The cleavage of pre-rRNA may constitute another checkmark that is required to allow snR30 dissociation. Accordingly, the absence of Utp23 may cause snR30 to remain bound to the SSU processome by inhibiting the pre-rRNA processing checkmark.

### 4.2. Timing of the snR30 snoRNP Release

Our knowledge about snR30 is limited by the fact that this snoRNA co-transcriptionally associates and dissociates from pre-rRNA already before the SSU processome is fully assembled. The timing of protein and snoRNA association with pre-rRNA, including snR30, was elegantly determined by analyzing stalled pre-ribosomal complexes assembled on truncated pre-rRNA [41]. Interestingly, snR30 can only be detected in the pre-ribosomal particle when about half of the central domain of 18S rRNA is already transcribed, including 100 nucleotides upstream of the rm2 motif. As expected, at the same time, H/ACA proteins Cbf5, Nop10, Gar1, and Nhp2 can be detected on the SSU processome, suggesting that snR30 is the H/ACA snoRNA that promotes most stable binding of the H/ACA proteins to the SSU processome. It is noteworthy that after transcription of snR30’s main interaction sites within 18S rRNA, namely the rm1 and rm2 sites, additional rRNA needs to be synthesized before snR30 can be stably detected. Possibly, this additional rRNA stretch stabilizes the ES6 where snR30 binds. Furthermore, transcription of a potential base-pairing between snR30 and a region in 18S rRNA downstream of rm2 (C3, vide supra) [34], which also encompasses the position of Utp23 crosslinking to pre-rRNA, might further stabilize snR30 binding to the SSU processome [87].

The snR30 snoRNP remains bound to the SSU processome until the 3′ major domain of 18S rRNA is transcribed [41]. This is a surprising finding as snR30 thus seems to leave the SSU processome before the A_2_ site in ITS1 is transcribed, although snR30 is crucial for facilitating A_2_ cleavage [26]. Further mechanistic research is required to reconcile these observations. Interestingly, the C/D box U14 RNP dissociates at the same time as snR30, whereas the U3 snoRNP remains present, forming the stable 90S particle observed by cryo-EM. The release of snR30, U14, and fourteen other factors, including proteins interacting with snR30 such as Cbf5, Utp23, Kri1, and Nop6, could be triggered by a conformational rearrangement during transcription of ITS1 [83,93,101]. According to these findings, it is possible that in particular snR30 and U14 may form a snoRNA complex within the SSU processome and constitute a functional unit.

## 5. Discussion

snR30/U17 is an unusual H/ACA guide RNA with a critical function during ribosome biogenesis that requires further mechanistic studies to be fully understood. With respect to ribonucleoprotein formation and synthesis, snR30/U17 behaves like a canonical H/ACA guide RNA. However, the essential function of snR30 differs greatly from the standard H/ACA modification RNAs as its binding orientation on the pre-rRNA and its effect on the pre-ribosome are entirely different. Thus, one of the most interesting challenges in the ribosome synthesis research field remains to uncover the molecular mechanism of the snR30 snoRNP in promoting pre-rRNA processing.

Research on snR30 is lagging behind other essential snoRNAs like U3 for several reasons, leaving many unanswered questions about its function. As for other essential RNAs, the cellular role of snR30 can only be assessed through transient knock-downs, mutational, and deletion studies, which all contributed to identify the functional elements of snR30 [74,84]. Additional crosslinking, pull-down, and mass spectrometry studies have helped to provide initial insight into the interactions of snR30 with the SSU processome [34,41]. However, these experimental approaches provide only limited insight into the functional mechanism of snR30 from ribosome binding to inducing some change in the ribosome to dissociating in a controlled manner, and we are therefore still missing a detailed description of snR30’s action on the SSU processome. Such understanding is particularly limited due to the transient interaction of snR30 with the maturing SSU processome as it has been reported to dissociate already before 18S rRNA transcription is complete [41]. This transient binding of snR30 with pre-rRNA has so far prevented us from obtaining high-resolution structural information on the interaction of this essential snoRNA with the SSU processome.

Two major hypotheses are proposed to explain the mechanistic role of snR30 within ribosome biogenesis. First, the protein recruitment hypothesis suggests that snR30 together with its associated proteins including Utp23 is essential to recruit and position additional factors on the SSU processome, ultimately helping Utp24 or another endonuclease to be correctly positioned for cleaving the pre-rRNA at sites A_0_, A_1_, and A_2_ (Figure 5). However, open questions remain regarding the details of the snR30-mediated interaction network and the timing of pre-rRNA processing. For example, we still do not know unambiguously the endonucleases responsible for A_0_, A_1_, and A_2_ cleavage, and it remains unclear how the timing of snR30 dissociation and the cleavage at the A_2_ site in the ITS1 are coordinated [36,41]. Second, the rRNA folding hypothesis indicates that snR30 may act as an RNA chaperone in promoting conformational changes within pre-rRNA that ultimately are propagated to enable positioning of endonucleases at the A_0_, A_1_, and A_2_ cleavage sites (Figure 5). Interestingly, in the cryo-EM structures of the SSU processome, expansion segment 6 is flexible and not folded into its mature conformation, such that it cannot be visualized [46]. If this complex represents a SSU processome structure after snR30 dissociation, then the function of snR30 could be to maintain the central domain of 18S rRNA in an unfolded state. Based on the secondary structure predictions of snR30 bound to 18S rRNA, it is equally conceivable that snR30 induces an alternative conformation in ES6. Both the un- or the re-folding of the 18S rRNA is likely a critical function of snR30, but in the absence of structural information, many questions regarding the impact of snR30 on SSU processome conformation remain.

In addition to our lack of understanding as to how snR30 promotes pre-rRNA processing, we also have limited knowledge on the timing, control, and mechanism of how the snR30 RNP is released from the SSU processome. Interestingly, snR30 requires two helicases, Has1 and Rok1, as well as Utp23 to dissociate from the SSU processome [93]. These three proteins may cooperate to signal a checkpoint that snR30 has completed its function before it dissociates. The dissociation of snR30 may be further coupled to progress in pre-rRNA transcription. In general, the finding that many factors are essential for the release of snR30 indicates that snR30 is responsible for an important step in ribosome maturation and that early dissociation of snR30 would abrogate small subunit formation. However, it remains to be uncovered how the SSU processome senses that snR30’s function is fulfilled such that it can be released and subsequent steps in ribosome biogenesis can occur.

In conclusion, snR30 is a highly interesting, still poorly understood snoRNA with critical function in ribosome biogenesis. This RNA, which has been hypothesized to be the ancestor to all H/ACA guide RNAs [102], is often overshadowed by the highly studied U3 RNA, but warrants further investigations to shed light not only on snR30, but also on critical steps during early ribosome synthesis. To answer the many open questions regarding snR30, we will need to apply a combination of traditional and innovative methods. Obviously, it will be most exciting to visualize snR30 bound to the SSU processome by cryo-EM. Significant challenges to purify a stable complex of an early ribosomal intermediate with snR30 will have to be overcome to reach this goal. Complementing such structural studies, now is the time to also utilize the power of in vitro studies to help elucidate the mysteries around snR30. Since the purification of active yeast H/ACA snoRNPs has been established, this route seems feasible to provide quantitative information on snR30’s RNA and protein interactions, to generate kinetic information on its impact on pre-rRNA and to ultimately generate mechanistic information [62,103]. Generally, solving the puzzle of snR30’s function will only be possible when we develop a thorough understanding on rRNA folding and transient rRNA–snoRNA interactions during ribosome synthesis, which will require innovative experimental approaches. Based on the rapid and stunning progress in uncovering the mechanism of ribosome formation in the past years, we predict that the next years will yield interesting insights into the role of snR30 mediating pre-rRNA processing and folding.

## Figures and Tables

**Figure 1 cells-09-02195-f001:**
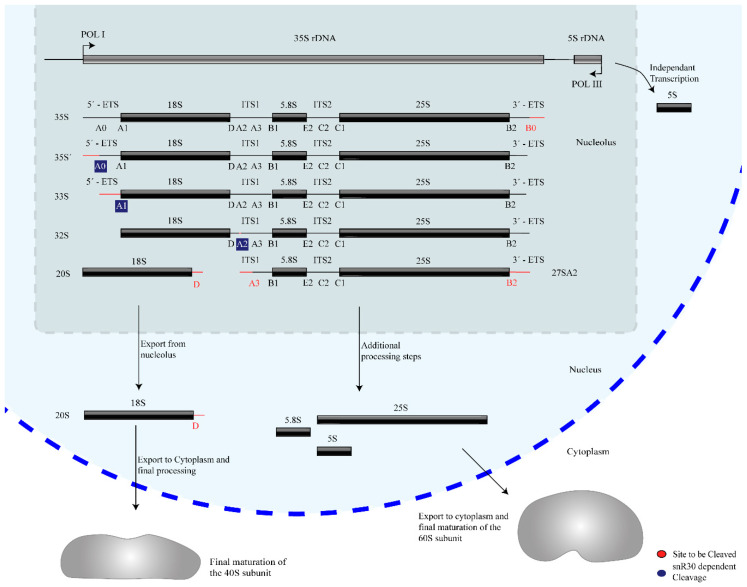
Overview of rRNA processing in yeast. The mature 18S, 5.8S, and 25S rRNAs are generated from a 35S pre-rRNA precursor through several nucleolytic cleavage events, shown here generating characteristic pre-rRNA intermediates. For each step, the subsequent site(s) of cleavage is highlighted in red. snR30 is essential for cleavage at sites A_0_, A_1_, and A_2_, which are highlighted in blue.

**Figure 3 cells-09-02195-f003:**
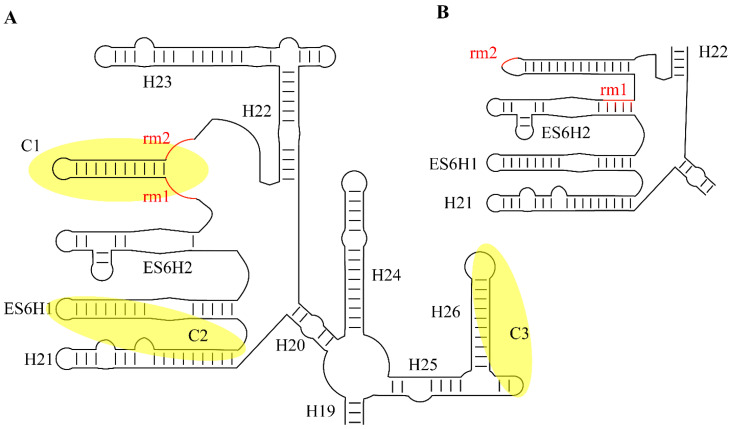
Conformational changes within ES6 of the 18S rRNA upon interaction with snR30. (**A**) Predicted rRNA conformation while snR30 base-pairs to the sequence elements rm1 and rm2 (red, crosslinking site C1) of the SSU processome. In addition, the crosslinking sites C2 and C3 are highlighted in yellow that were observed for *S. cerevisiae* snR30 (compare to Figure 2 for the corresponding crosslinking sites in snR30). (**B**) rRNA secondary structure of ES6 as visualized in cryo-EM models of the mature ribosome showing the different base-pairing in absence of snR30 [86].

**Figure 4 cells-09-02195-f004:**
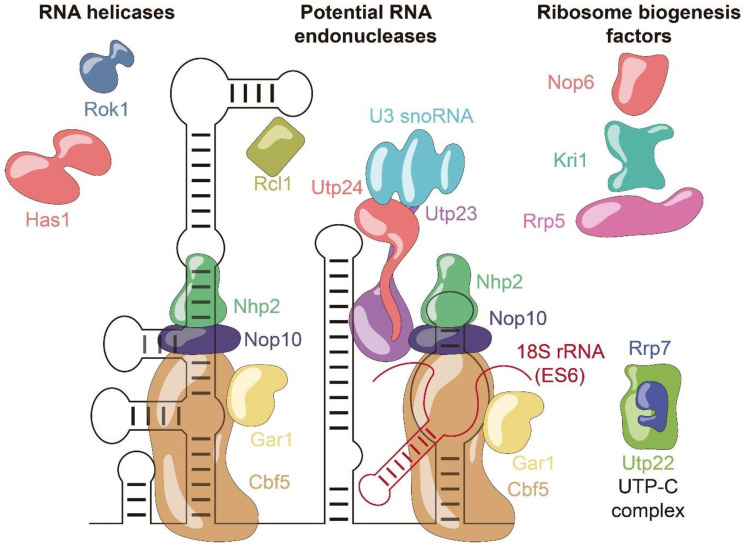
Schematic representation of the snR30 protein interaction network. snR30 snoRNA and its binding partners relevant to ribosome biogenesis are sorted and highlighted by different colors according to protein function. snR30 RNA is displayed in black, and pre-rRNA is in red.

**Figure 5 cells-09-02195-f005:**
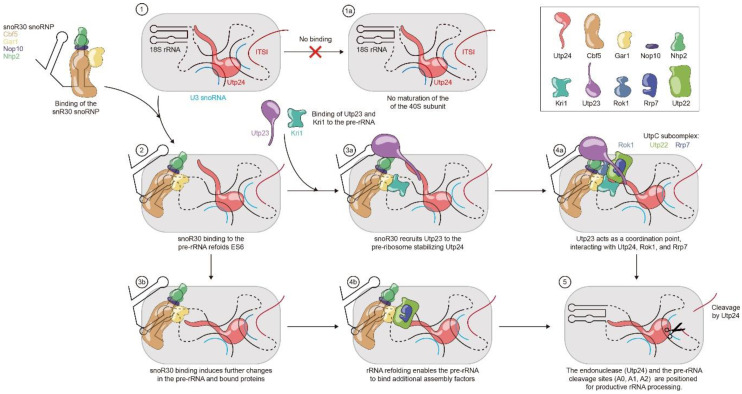
Putative molecular mechanisms of snR30 during ribosome biogenesis. A schematic of the SSU processome is shown as a light gray background with Utp24 (cyan) bound to the 18S rRNA and U3 snoRNA (**1**). In the absence of snR30, no processing occurs (**1a**). Base-pairing of snR30 to the rm1 and rm2 sites in expansion segment 6 (ES6) of 18S rRNA causes the two hairpins in ES6 to unfold and to instead refold into a single hairpin (**2**). Once bound to the SSU processome, snR30 likely recruits the proteins Utp23 and Kri1 to the SSU processome (**3a**). Utp23 may act as a coordination point interacting with several other proteins such as Rrp7, Rok1, and the endonuclease Utp24 (**4a**). These interactions could lead to a stabilization and possible reorganization of the SSU processome, allowing processing to occur, e.g., by bringing the endonuclease(s) (Utp24 or Rcl1) to the pre-rRNA cleavage sites (**5**). Alternatively, or in addition, the rearrangement of ES6 upon snR30 binding as well as other folding and unfolding events in rRNA caused by snR30 may induce conformational changes throughout the 18S pre-rRNA (**3b**). This could lead to repositioning of already bound factors like Utp24 and to the recruitment of essential factors like the UtpC subcomplex (**4b**). Ultimately, these conformational changes in the SSU processome correctly position the endonucleases relative to the pre-rRNA cleavage site within the SSU processome (**5**). snR30 and the 18S section of the pre-rRNA is black, the ITS1 of the pre-rRNA is red, and U3 snoRNA is depicted in light blue.

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
