# Peer review of "snR30/U17 Small Nucleolar Ribonucleoprotein: A Critical Player during Ribosome Biogenesis"

_cells, 2020, doi:10.3390/cells9102195_

Round 1

Reviewer 1 Report

In this manuscript, Vos and Kothe provide a comprehensive overview on the role and biological function(s) of snR30, an essential H/ACA snoRNA required for the early steps of ribosome biogenesis.

First, Vos and Kothe properly delineate the biological context where snRN30 is involved in, and accordingly summarize key aspects of ribosome and snoRNP biogenesis. Finally, based on existing literature Vos and Kothe propose a couple of interesting hypotheses regarding the possible mechanism(s) of action of snR30.

Overall, the review is well written, scholarly balanced, and provide an up-to-date view on the still poorly understood snR30 biology.

I have only few comments/suggestions that the authors may consider.

  1. Figure 1: The colour combination (red/blue) to highlight some of the rRNA sites is difficult to read. Font size and/or style and/or colour combination should be modified.
  2. The cartoon drawing in Figure 4 and 5, is in principle correct and simple enough to be easily understood, however, it would be more appealing if aesthetically made more attractive.
  3. Figure 5, legend: The “90S processome” is probably not the most appropriate nomenclature unless the authors better define the meaning of it. Especially, since there is already some nomenclature ambiguity and confusion in the field. In general, the different terminology used are not introduced in this review, which is fine in principle, but I would avoid introducing a new one combining previous nomenclatures without explanation and referring to the previous ones, e.g. SSU processome (as defined by Baserga), 90S (as defined by Planta/Warner containing 35S rRNA) and 90S (as defined by Hurt).
  4. Table 1: I think one should read Tetrahymena thermophila and not thermophilus. Please correct Table and legend as well and any further occurrence in the text, if any.
  5. P3 lane 75: to be correct this sentence should refer to both type of SSU rRNA i.e. 16S/18S rRNA or more generally to SSU rRNA.
  6. P3 lane 79: “… aberrant 23S pre-rRNA …” In many previous studies the 23S rRNA is often described as an aberrant precursor, however, whether the 23S pre-rRNA is only a non-productive intermediate is rather difficult to say and is experimentally not clear, e.g. PMID: 28282370 and PMID: 27036125.
  7. P4 lane 125: emphasis should be made that these conclusions were not gained from yeast (not speaking that these data are in contradiction with most of our knowledge gathered in prokaryotic and eukaryotic model systems, so far).
  8. P9 “3.1 Site-Specific Binding to the 18S rRNA”: The authors should more comprehensively cover/discuss snR30 cross-linking sites on the 18S rRNA see PMID: 24947498. This part could be also enhanced by summarizing these findings graphically and highlighting them into the different structural context available: i.e. 2D 16S rRNA map and 3D (pre-) mature small ribosomal subunits. This would help the readers to appreciate and better visualize the putative “topological surrounding” and “region of action” of snR30, and how these regions might locally develop/change during SSU maturation.
  9. P10 lane 310: it should be proposed to, to is missing.
  10. P11 “3.3 Hypothetical functions of snR30 during ribosome formation”: My feeling is that the authors could also speculate on at least one additional hypothesis which does not seem to be explicitly covered by the two currently proposed hypotheses. For example, snR30 could avoid the formation of premature folding intermediates (of ES6 itself and/or additional regions).
  11. P16 lane 480-83. To avoid confusion, the snR30 and mentioned factors  are not released from the ribosome as currently written, but from the nascent (small) ribosomal subunit, or pre-ribosomal subunit.

Author Response

We thank you for critically reading our manuscript about the essential H/ACA snoRNA snR30 and for providing constructive feedback to further improve this review. We appreciate that all reviewers consider this an interesting, comprehensive and timely article.

Based on your comments, we have made the following changes:

  1. Figure 1: The colour combination (red/blue) to highlight some of the rRNA sites is difficult to read. Font size and/or style and/or colour combination should be modified.
    • We changed the font to white to increase the contrast to the blue background.
  2. The cartoon drawing in Figure 4 and 5, is in principle correct and simple enough to be easily understood, however, it would be more appealing if aesthetically made more attractive.
    • We have recruited the help of a professional designer to render Figure 4, Figure 5 and the graphical abstract aesthetically more attractive.
  3. Figure 5, legend: The “90S processome” is probably not the most appropriate nomenclature unless the authors better define the meaning of it. Especially, since there is already some nomenclature ambiguity and confusion in the field. In general, the different terminology used are not introduced in this review, which is fine in principle, but I would avoid introducing a new one combining previous nomenclatures without explanation and referring to the previous ones, e.g. SSU processome (as defined by Baserga), 90S (as defined by Planta/Warner containing 35S rRNA) and 90S (as defined by Hurt).
    • Thank you. To minimize ambiguity, we now use the term “SSU processome” predominantly throughout the entire manuscript including the legend of Figure 5.
  4. Table 1: I think one should read Tetrahymena thermophila and not thermophilus. Please correct Table and legend as well and any further occurrence in the text, if any.
    • Table 1; p9, line 262 & 266: we have corrected the name to Tetrahymena thermophila.
  5. P3 lane 75: to be correct this sentence should refer to both type of SSU rRNA i.e. 16S/18S rRNA or more generally to SSU rRNA.
    • p3, line 74 & 75: We agree and changed the text to SSU rRNA.
  6. P3 lane 79: “… aberrant 23S pre-rRNA …” In many previous studies the 23S rRNA is often described as an aberrant precursor, however, whether the 23S pre-rRNA is only a non-productive intermediate is rather difficult to say and is experimentally not clear, e.g. PMID: 28282370 and PMID: 27036125.
    • p3, line 19: we have removed the term “aberrant”.
  7. P4 lane 125: emphasis should be made that these conclusions were not gained from yeast (not speaking that these data are in contradiction with most of our knowledge gathered in prokaryotic and eukaryotic model systems, so far).
    • p4, line 125ff: we have added that this study was conducted in thermophilum and also added a sentence of caution that it is not know whether the conclusions are generalizable to other organisms.
  8. P9 “3.1 Site-Specific Binding to the 18S rRNA”: The authors should more comprehensively cover/discuss snR30 cross-linking sites on the 18S rRNA see PMID: 24947498. This part could be also enhanced by summarizing these findings graphically and highlighting them into the different structural context available: i.e. 2D 16S rRNA map and 3D (pre-) mature small ribosomal subunits. This would help the readers to appreciate and better visualize the putative “topological surrounding” and “region of action” of snR30, and how these regions might locally develop/change during SSU maturation.
    • p9, Section 3.1: building on the information on the crosslinking sites of snR30 on the 18S rRNA as already introduced on page 7 & 8, line 241 – 251, we have added sentences in this section to discuss the proximity of the crosslinking sites in the 2D of 18S rRNA and the 3D structure of the SSU processome (p9, line 289 – 298). Moreover, we have up-dated Figures 3A to include and highlight the crosslinking sites C1, C2 and C3 in the 2D structure of 18S rRNA.
  9. P10 lane 310: it should be proposed to, to is missing.
    • p10, now line 330: Thank you! The missing “to” has been added.
  10. P11 “3.3 Hypothetical functions of snR30 during ribosome formation”: My feeling is that the authors could also speculate on at least one additional hypothesis which does not seem to be explicitly covered by the two currently proposed hypotheses. For example, snR30 could avoid the formation of premature folding intermediates (of ES6 itself and/or additional regions).
    • p12, line 372ff as well as p14, lin3 400 (Figure 5 caption): We could not agree more with the reviewer! Therefore, we have clarified that the discussed conformational changes may include both folding events in 18S rRNA as well as the unfolding of prematurely formed structures.
  11. P16 lane 480-83. To avoid confusion, the snR30 and mentioned factors are not released from the ribosome as currently written, but from the nascent (small) ribosomal subunit, or pre-ribosomal subunit.
    • p16, line 507ff: Thank you! We have replaced the word ribosome with SSU processome.

Reviewer 2 Report

The review provides a comprehensive and interesting look at snR30/U17 snoRNA, including expression, function, and protein interactions. The authors provide some hypothesized future directions for readers to consider.  There are some suggestions for the authors to consider.

  1. In Figure 1, it is a little difficult to read the red font with the blue boxes. I would suggest changing to white font, or perhaps a darker color blue for the box itself.
  2. In Table 2, would it be possible in indicate more vertebrate information, and specifically more on the host gene that contains the snoRNA? Should common organism names be included in Table 2 as well?  Related to Table 2, could information on the phylogenetic conservation be included as a figure—perhaps with each of the sequences from the Table 2 data lined up with IUPAC conservation codes.
  3. Figure 4 is very useful, but the figure legend should indicate that snR30 RNA is in black and the 18s rRNA is in red (rather than in the figure itself). Similar comment for Figure 5.

Author Response

We thank you for critically reading our manuscript about the essential H/ACA snoRNA snR30 and for providing constructive feedback to further improve this review. We appreciate that all reviewers consider this an interesting, comprehensive and timely article.

Based on your comments, we have made the following changes:

  1. In Figure 1, it is a little difficult to read the red font with the blue boxes. I would suggest changing to white font, or perhaps a darker color blue for the box itself.
  • Figure 1: We have changed the font to white as suggested.
  1. In Table 2, would it be possible in indicate more vertebrate information, and specifically more on the host gene that contains the snoRNA? Should common organism names be included in Table 2 as well?  Related to Table 2, could information on the phylogenetic conservation be included as a figure—perhaps with each of the sequences from the Table 2 data lined up with IUPAC conservation codes.
    • Table 1: We have added information on mouse, drosophila, and zebrafish in the table as widely used model organisms, and we also include now the common organism names. Thank you for these suggestions! As this review is already quite long, and as snR30/U17 varies significantly in sequence and structure as we discuss using the yeast and human example, we think that it is beyond the scope of this review to also include a genetic and phylogenetic analysis. Hopefully, Table 2 provides sufficient information for an interested reader to further explore this topic.
  2. Figure 4 is very useful, but the figure legend should indicate that snR30 RNA is in black and the 18S rRNA is in red (rather than in the figure itself). Similar comment for Figure 5.
    • Figure 4 (page 12, line 363) and Figure 5 (page 14, line 406): the figure captions have been expanded to include a description of the snR30 and pre-rRNA color.

Reviewer 3 Report

The role of snoRNAs during ribosome biogenesis is of extreme importance, since it is a complex, difficult and multilevel process, therefore such systematization of the existing data and careful criticism is of special importance. I like the text of this review a lot, the figures are ok (maybe a little bit too dark, especially 1 and 4). The discussion clarifies and presents the strenght of this elaboration.

Author Response

We thank you for critically reading our manuscript about the essential H/ACA snoRNA snR30 and for providing constructive feedback to further improve this review. We appreciate that all reviewers consider this an interesting, comprehensive and timely article.

Based on your comments, we have made the following changes:

  • We have adjusted the colors in Figure 1.
  • Figure 4 has been completely revised.